# Clinical outcomes with lower versus conventional dose polymyxin B regimens in dialysis dependent and non-dialysis patients with gram-negative sepsis: A real-world propensity-score matched cohort study

Asha K. Rajan[1], Vishal Shanbhag [ID][2], Vijayanarayana Kunhikatta[1], Ravindra Prabhu Attur[3], Beven Nelson[4], Varun Kumar S. G[4], Souvik Chaudhuri[2], Girish Thunga [ID][1]*

1 Department of Pharmacy Practice, Manipal College of Pharmaceutical Sciences, Manipal Academy of Higher Education, Manipal, Karnataka, India, 2 Department of Critical Care Medicine, Kasturba Medical College, Manipal Academy of Higher Education, Manipal, Karnataka, India, 3 Department of Nephrology, Kasturba Medical College, Manipal Academy of Higher Education, Manipal, Karnataka, India, 4 Department of Applied Statistics and Data Sciences, Prasanna School of Public Health, Manipal Academy of Higher Education, Manipal, Karnataka, India

* girish.thunga@manipal.edu

## Abstract

### Background

Polymyxin B remains a key treatment option for infections caused by multidrug-resistant gram-negative bacilli, particularly in critically ill patients. However, its optimal dosing strategy recommendation remains uncertain, especially in those undergoing renal replacement therapy. This study aimed to compare the clinical and microbiological outcomes of low, usual and high dose polymyxin B in a real-world ICU population.

### Methods

This 5-year retrospective cohort study included critically ill adult patients with gram-negative sepsis who received polymyxin B. Patients were categorized into low-, usual- and high-dose groups based on loading and total daily maintenance dose. Pairwise propensity score matching was performed to adjust for baseline differences. Primary outcome was 28-day all-cause mortality. Secondary outcomes included microbiological clearance, ventilator-free days, ICU-free days, and vasopressor-free days. Subgroup and sensitivity analyses were conducted, including within patients requiring dialysis. All the statistical analysis was performed using R software.

### Results

A total of 674 patients were included. After matching, usual-dose polymyxin B (61%) was associated with significantly higher 28-day mortality compared to the low-dose group (48.04%) (HR = 1.47; 95% CI:[1.11–1.95]; p = 0.007). Vasopressor, ventilator and

**Data availability statement:** All relevant data are within the manuscript and its supporting information files. The raw data sets of patients are available from the corresponding author upon request. Any additional data related to ethical details of the study are available from Kasturba Medical College and Kasturba Hospital Institutional Ethics Committee, email: iec.kmc@manipal.edu.

**Funding:** The author(s) received no specific funding for this work.

**Competing interests:** The authors have declared that no competing interest exist.

**Abbreviations**: AKI: Acute Kidney Injury, APACHE II: Acute physiology and chronic health evaluation, ARDS: acute respiratory distress syndrome, CCI: Charlson comorbidity index, DIC: Disseminated intravascular coagulation, MDR: multi-drug resistant, MODS: Multi organ dysfunction syndrome, RRT: Renal Replacement Therapy, SOFA: Sequential organ failure assessment, TDM: Therapeutic Drug Monitoring.

ICU-free days were also significantly higher in the low-dose group were compared to the other groups. No significant survival advantage was observed with high-dose regimens. Among dialysis-dependent patients (n = 254), mortality did not differ significantly across dosing groups, though microbiological clearance was better with low dosing. Sensitivity and subgroup analysis also supported the results to be robust.

## Conclusion

Low dose polymyxin B regimens were associated with lower mortality and comparable clinical outcomes compared to higher doses and may be feasible in critically ill patients with renal impairment. However, these findings should be interpreted cautiously given the observational design and residual confounding, warranting confirmation in future randomized trials.

---

## 1. Introduction

The increasing prevalence of multi-drug-resistant (MDR) bacteria is associated with high attributable mortality, primarily due to the limited availability of effective therapeutic agents [1]. Among carbapenem-resistant organisms, the widespread antimicrobial resistance severely restricts treatment options, posing a significant clinical challenge [2]. Although the polymyxin class of antibiotics- namely polymyxin E (colistin) and polymyxin B- fell out of favour shortly after their introduction in the 1950s due to dose-limiting toxicities, their clinical use has resurged in recent years in response to the growing burden of infections caused by MDR and extensively drug-resistant (XDR) Gram-negative pathogens [3]. Notably, polymyxins remain one of the few antibiotic classes with consistent activity against carbapenem-resistant strains of *K. pneumoniae* and *A. baumannii* [4].

In many clinical settings, polymyxin B is preferred over colistin due to its more predictable pharmacokinetics and rapid bactericidal activity [5]. However, several challenges are encountered in the use of polymyxins in patients with renal impairment, especially given the variable clinical behaviour observed in sepsis and septic shock [6]. Moreover, discrepancies persist regarding optimal dosing strategies in patients with chronic kidney disease (CKD) or acute kidney injury (AKI), with minimal guidance on supplemental dosing in those undergoing renal replacement therapy (RRT).

Current clinical guidelines recommend a weight-based dosing regimen of 1.25–1.5 mg/kg every 12 hours for polymyxin B, aiming to achieve a target area under the concentration-time curve over 24 hours ($AUC_{0-24}$) at a steady-state plasma concentrations of 2–4 mg/L [7]. Nonetheless, the optimal dosing strategy for polymyxin B remains controversial [8]. Furthermore, therapeutic drug monitoring (TDM) of polymyxin B is not widely implemented in clinical practice. This has led to substantial interindividual variability in drug concentrations among critically ill patients, which may contribute to suboptimal clinical outcomes [9].

Studies have also reported that the polymyxin B doses administered are often lower than those recommended by guidelines, with clinicians frequently opting for

fixed doses of 5,00,000U (50 mg) or 7,50,000U (75 mg) every 12 hours [10]. While this approach may not account for total body weight, it is easier to implement and reduce drug wastage. Additionally, some patients do not receive an initial loading dose of polymyxin B, which may further compromise therapeutic efficacy [11,12].

Currently, there is a lack of robust clinical evidence and consensus guidelines regarding the optimal dosing of polymyxin B, particularly in septic patients undergoing RRT. Therefore, we aimed to evaluate the effectiveness of various polymyxin B dosing strategies on clinical outcomes and to determine an appropriate dosing regimen in critically ill patients with sepsis. The insights gained aim to support the development of rational dosing strategies that ensure adequate drug exposure for treating infections caused by polymyxin-susceptible pathogens.

## 2. Materials and methods

### 2.1. Ethical statement

This study adhered to the principles of the Declaration of Helsinki and received approval from the Institutional Ethics Committee [IEC approval no: 388/2022]. Given its retrospective design, the requirement for informed consent was waived. The medical records used in this study were accessed from 07/03/2023–30/11/2024. All data were retrieved from the institutional electronic health record system by the study investigators. During data collection, the investigators have access to patient identifiers to ensure accurate matching and extraction of relevant variables. However, all data were anonymized immediately after extraction, and no identifiable information was used in the final analysis. The dataset used for statistical evaluation contained only de-identified records in compliance with ethical and data protection standards.

### 2.2. Study design, setting, and patient population

We conducted a retrospective, observational cohort study at a tertiary-care hospital, including all adult patients (≥ 18 years) diagnosed with sepsis or septic shock (ICD.10 code: A41.1-41.9) who received polymyxin B therapy alone or in combination with colistin (intravenous)/ inhalational), between January 2018 and December 2022 in the ICU or High Dependency Unit (HDU). Exclusion criteria were: (1) Polymyxin B treatment <3 days, (2) colistin monotherapy, (3) paediatrics or neonates, (4) incomplete medical records, or (5) discharge against medical advice. Data were extracted from medical records and complied into a comprehensive study database. Fig 1 presents the overall methodological workflow.

### 2.3. Definitions

Sepsis is a life-threatening organ dysfunction arising from a dysregulated host response to infection. Septic shock is a subset of sepsis characterized by profound circulatory and metabolic abnormalities associated with higher mortality risk [13,14]. Multidrug resistance was defined as the non-susceptibility to at least one antibiotic agent in ≥3 antimicrobial classes [15]. Microbiological clearance was defined as the absence of the baseline pathogen in follow-up cultures obtained ≥72 hours after starting polymyxin B therapy. with no subsequent isolation of the same organism. Patients without repeat cultures were conservatively classified as "not cleared". This stringent definition was chosen to minimize misclassification, but it may underestimate clearance compared with more permissive criteria. Minimum inhibitory concentrations were not uniformly available because susceptibility testing during the study period was routinely performed using disc diffusion methods, and MIC determination was only conducted selectively. Consequently, we were unable to assess whether treatment groups differed in the distribution of MIC values or to evaluate exposure-response relationships.

AKI is the acute decline in renal function, typically reflected by reduced glomerular filtration rate (GFR) and urine output [16]. Although AKI events were extracted from clinical records, standardized adjudication using KDIGO criteria was not feasible because serial serum creatinine measurements and urine output data were inconsistently available at required time points, particularly in patients with early initiation of renal replacement therapy. ARDS is the sudden-onset, diffuse inflammatory lung injury with impaired oxygenation and pulmonary infiltrates [17]. Severe encephalopathy is the diffuse cerebral dysfunction presenting as delirium, acute confusional state, or coma [18]. Cardiac complications were defined

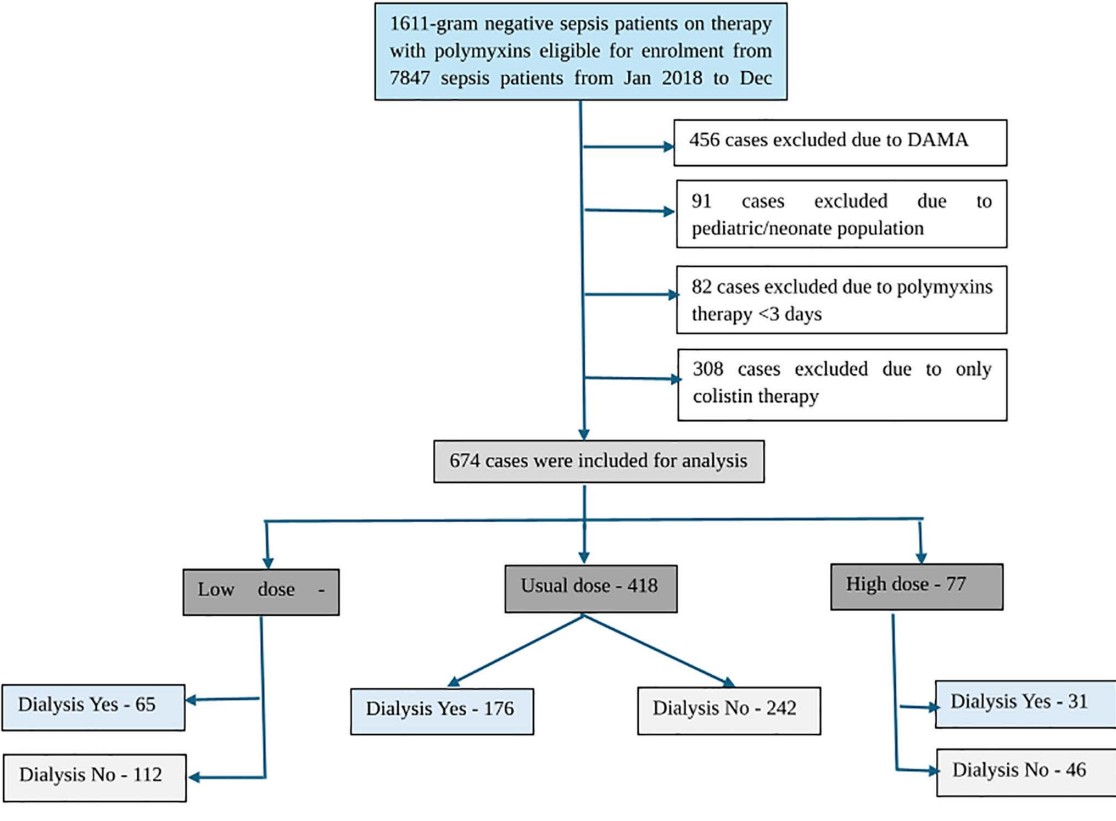

**Fig 1. Workflow of the study.**

as a new onset of arrhythmias, atrial fibrillation, or myocardial infarction. Disease severity was assessed using the Acute Physiology and Chronic Health Evaluation II (APACHE II), Sequential Organ Failure Assessment (SOFA), and Charlson Comorbidity Index (CCI) scores.

## 2.4. Exposure and follow-up

Patients were followed from the initiation of polymyxin B therapy until hospital discharge or death. Cohort entry was defined as the date of initiation of polymyxin B therapy for sepsis. Clinical and microbiological data were extracted retrospectively from electronic medical records. Microbiological assessment spanned from the first culture obtained at admission through the end of therapy, with cultures ordered per clinician discretion. We compared pre-and post-therapy cultures to evaluate polymyxin B's effect on microbiological clearance. When clinical documentation was inconsistent (e.g., Infection source not specified or recorded differently across notes), we used the consensus of two independent reviewers (clinical pharmacist and critical care physician) to adjudicate the most likely source or classified it as "undocumented" if no consensus could be reached. Dialysis modality was recorded from patient charts and categorized as sustained low efficiency dialysis (SLED), continuous renal replacement therapy (CRRT) or intermittent hemodialysis (IHD).

Polymyxin B dosing was at the treating physician's discretion, guided by available evidence but without formal adjustment for renal replacement therapy (RRT). There was no adjustment of polymyxin B dosing in our setting in patients with or without renal replacement therapy with the maintenance doses ranging from 5,00,000U to 20,00,000U. Dosing categories were defined according to the prescribing practices at our centre, where doses are routinely documented in lakh

units (LU) rather than weight-based regimens. For categorization, we adopted thresholds reflecting the most frequently observed patterns in clinical practice. We categorized polymyxin B dosing into three strategies- "low", "usual" and "high"- with specific loading and maintenance regimens as defined in Table 1. Although these definitions differ from international guideline recommendations, [weight-based dosing (LD 2.0–2.5 mg/Kg; MD 1.25–1.5 mg/kg q12h)], they are consistent with prior observational studies that have reported fixed-dose regimens in similar settings [10]. Unfortunately, patient-level body weight was not consistently recorded in our dataset, preventing a formal mg/kg-based analysis. Patients were classified according to the predominant dosing regimen used during therapy. In cases where loading and maintenance doses differed (e.g., High loading dose but usual maintenance dose), categorization was based on the maintenance regimen, as this reflects ongoing exposure.

Polymyxin B dosing in our centre is routinely prescribed and documented in fixed lakh-unit doses rather than as weight-based regimens. Accurate body weight was not consistently available for critically ill ICU patients due to clinical instability and logistical constraints, precluding recalculation of doses on a mg/kg basis. Consequently, dosing categories ('low', 'usual' and 'high') were derived from centre-specific prescribing patterns and reflect real-world practice rather than guideline-recommended, weight-adjusted dosing strategies. This approach captures actual clinical decision-making in our setting but limits direct pharmacokinetic interpretation and comparison with international dosing recommendations.

## 2.5. Variables

We selected candidate variables based on clinical relevance and prior evidence in sepsis. These included: Demographics: age, gender, comorbidities, pre-admission clinical presentation, vital signs, GCS, and dialysis requirement. Severity scores such as APACHE II, SOFA and CCI. Microbiological details such as resistance pattern and culture results from various infection sites, pre- and post-polymyxin B therapy.

Empirical antibiotic therapy within 24 hours and duration >3 days, polymyxin B loading and maintenance doses (continuous, later categorized into "low", "usual", and "high"), two most common concomitant antibiotics. Complications noted down included incidence of septic shock, AKI, multi-organ dysfunction syndrome, acute respiratory distress syndrome (ARDS), disseminated intravascular coagulation, pneumothorax, thromboembolism, cardiac events, and if more than one complication exists.

Outcomes: The primary outcomes were 28-day all-cause mortality after initiating therapy with polymyxin B (death or discharge with clinical improvement), whichever occurred first. Secondary outcomes included microbiological clearance, vasopressor free days, ventilator free days and ICU free days after initiating on therapy with polymyxin B along with total duration of polymyxin B therapy. All available data on the safety and effectiveness parameters of polymyxin B were retrieved from patient's medical records and systematically analyzed.

## 2.6. Management of missing data

Variables with >20% missing values were excluded. For variables with <20% missingness, we applied median imputation; extreme or implausible values were treated as missing prior to imputation. A full list of imputed variables is included in S1 Table.

Table 1. Dosing strategies of polymyxin B adapted in our study [10].

| Dosing strategy | Loading dose | Maintenance dose |
|---|---|---|
| Low dose | <150mg/<15LU | ≤150 mg/≤15LU |
| Usual dose | 150mg/15LU | 150mg/15LU |
| High dose | >150mg/>15LU | >150mg/15LU |

*LU- lakh units.

## 2.7. Statistical analysis

Given the retrospective design of this study and the potential for confounding, we employed robust statistical techniques to mitigate bias. Propensity score matching was performed using clinically meaningful covariates, including APACHE II, SOFA, and CCI score, dialysis status, and relevant baseline demographics, to ensure comparability between dosing groups [19]. Patients were stratified into three dosing arms- "low", "usual", and "high". Categorical variables are summarized as frequencies (percentages) with between-group comparisons by chi-square test. Continuous variables are reported as mean ± standard deviation (SD) or median ± interquartile range (IQR), depending on normality assessed via Kolmogorov-Smirnov and Shapiro-Wilk tests, with group comparisons conducted using the Kruskal-Wallis tests. To detect multicollinearity among covariates, we calculated variance inflation factors (VIF) and tolerance (T) values, with VIF > 5 and T < 0.1 indicating multicollinearity.

For our primary analysis, we considered all established prognostic factors influencing polymyxin B dosing- including age, sex, comorbidities, prior hospital admissions, transfer from external facilities, and history of carbapenem resistance or therapy- and incorporated APACHE II score, SOFA score, CCI and dialysis requirement at the time of polymyxin B initiation into the propensity model. The propensity score, reflecting each patient's probability of assignment to a given dosing strategy (LD, UD or HD), facilitated pairwise 1:1 nearest-neighbour matching without replacement ("LD vs UD", "UD vs HD", LD vs HD") with a caliper of 0.1 SD of the logit of the score. This caliper was chosen based on established recommendations from simulation studies by Austin et al., [19] which suggest that a 0.1 SD caliper achieve optimal balance while minimizing bias and variance in most clinical datasets.

We assessed covariate balance before and after matching using standardized mean differences. Because the composite severity scores encompass demographic, laboratory and clinical presentation data, we did not individually match on these variables. Detailed balance diagnostics are presented in S2 Table. While TDM data were unavailable, stratification by dialysis modality and detailed documentation of dosing regimens allowed us to account for differences in polymyxin B exposure indirectly. The same method of matching was used to understand the better dosing strategies in the dialysis requiring cohort using the covariates namely, APACHE II, SOFA and CCI scores.

After matching, we applied Cox proportional hazard regression to estimate the effect of dosing strategy on 28-day mortality and logistic regression to evaluate associations with microbiological clearance. Mann-Whitney U test compared vasopressor-, ventilator- and ICU -free days among dosing groups. We conducted a sensitivity analysis using various designs of matching namely, 1:2 and 1:3 and performed subgroup analyses by sex (male and female) for the primary endpoint of 28-day mortality to check the robustness of the results. Kaplan-Meier survival curves with log rank tests illustrated differences in time-to-event (from polymyxin B initiation to death or discharge) across pairwise comparisons. Statistical significance was defined as two-tailed P ≤ 0.05. All analyses were conducted using the statistical software package R version 4.1.3 (R project for statistical computing, Vienna, Austria).

## 3. Results

### 3.1. Patient characteristics

A total of 1611 gram-negative sepsis patients on therapy with polymyxin B were eligible for enrolment. A flow diagram showing the numbers of cases and reasons for exclusion is shown in Fig 1. A total of 674 patients were included in the analysis and categorized into low-dose (n = 179), usual-dose (n = 418) and high-dose (n = 77) polymyxin B treatment groups. The median age was comparable across groups (p = 0.076), although a statistically significant difference was noted within the distribution of age categories (p = 0.011). Male predominance was observed in all groups, with the highest proportion in the high-dose group (70.13%). A significantly higher proportion of patients had chronic respiratory failure (BA, COPD, ILD) which was more frequent in the high-dose group (p = 0.003). Severity scores, including CCI, APACHE II and SOFA score were similar between groups. Among patients requiring dialysis, modalities included SLED (n = 93), CRRT (n = 9), both HD and SLED (n = 141) and HD (n = 29). Their distribution across

dosing groups is shown in Table 2. No significant imbalance in dialysis modality was observed between groups. Also, no significant differences were observed between groups for any baseline covariate after matching (p > 0.05), indicating adequate balance (S3 Table).

### 3.2. Clinical outcomes in the included patients after polymyxin B therapy before matching

Mean loading dose of polymyxin B in the included cohort was 13.97 ± 2.92LU followed by a mean maintenance dose of 7.25 ± 1.47LU. The mortality rate was highest in the high-dose group (62.3%), followed by the usual-dose group (61%) and the lowest in the low-dose group (48%), with a significant difference noted (p = 0.009). The length of hospital stay was longest in the low-dose group, compared to the others (p = 0.004). However, no statistically significant difference was observed in septic shock or sepsis-related AKI across groups. These AKI events reflect clinician documented diagnoses rather than standardized KDIGO-defined nephrotoxicity and should therefore be interpreted descriptively rather than as a formal safety comparison across dosing groups.

Regarding complications, ARDS was most frequent in the high-dose group (18.2%), followed by the usual and low dose group. DIC and MODS were also more prevalent in the high-dose group compared to the others. Cardiac complications and severe encephalopathy occurred at similar frequencies across all dosing groups. Outcomes in the overall cohort (pre-matching and post-matching) are summarized in Tables 3 and 4. We plotted bar charts representing microbiological cultures from all the prominent source of culture pre-and post-polymyxin B therapy (S1 Fig.).

### 3.3. Low dosing strategy was associated with a better clinical outcome among all the included cohort patients after polymyxin B therapy

After propensity score matching, all included covariates demonstrated good balance with standardized mean differences below 0.1 across all pairwise comparisons, confirming the adequacy of the matching procedure. The comparisons across polymyxin B dosing strategies revealed notable difference in clinical outcomes, particularly between the low and usual-dose groups. 28-day mortality, assessed using Cox proportional hazards modeling, was significantly higher in the usual-dose group compared to the low-dose group (HR = 1.47;95%CI:[1.11–1.95];p = 0.007) indicating a 47% increased risk of death within 28 days among patients receiving usual dose. Conversely, there was no significant difference in mortality between the other two comparison groups.

Microbiological clearance was comparable across all dosing group, with no statistically significant differences observed in any pairwise comparison. Microbiological clearance was very low across all dosing groups (12–15%), which likely reflects the severity of the infections and the inherent difficulty of achieving eradication in critically ill patients. In many cases, polymyxin B was initiated as salvage therapy at an advanced stage of sepsis, by which time microbiological clearance was rarely observed. Ventilator-free days was significantly higher in the low-dose group compared to the usual-dose group [9(0–106) vs 3(0–76);p = 0.00002]. Similarly, ICU-free days were significantly high in the low-dose group compared to the usual-dose group [9(0–111) vs 4(0–77);p = 0.003]. Also, vasopressor-free days was significantly high in the low-dose group compared to the usual-dose group [3(0–46) vs 4(0–50);p = 0.005]. This may reflect differences in illness severity or responsiveness to therapy. No significant differences in vasopressor-free days, ventilator-free days and ICU free days were observed between the other two pairwise group comparisons. Outcomes in the propensity score-matched cohort are shown in Table 5.

Kaplan-Meier survival analysis revealed significant difference in 28-day mortality among the polymyxin B dosing strategies. Patients receiving the low-dose regimen demonstrated a significantly better survival probability compared to those on the usual dose (log rank p = 0.0075). In contrast, no statistically significant difference was observed between the low and high-dose group (log-rank p = 0.33) and usual and high-dose group (log-rank p = 0.96). Fig 2 illustrates the Kaplan-Meier plots of survival analysis.

**Table 2. Baseline study characteristics of the patients.**

| Variables | Low dose (n = 179) | Usual dose (n = 418) | High dose (n = 77) | p value |
|---|---|---|---|---|
| Age | 56 (19-89) | 58 (19-89) | 52 (19-85) | 0.076* |
| Age category | | | | **0.011**** |
| 18-30: | 30 (16.76) | 38 (9.09) | 10 (12.99) | |
| 31-50: | 35 (19.55) | 98 (23.44) | 24 (31.17) | |
| 51-70: | 83 (48.72) | 213 (50.96) | 33 (42.86) | |
| 71-90: | 28 (15.64) | 69 (16.51) | 10 (12.99) | |
| >91: | 3(1.68) | – | – | |
| Gender | | | | **0.036**** |
| Male | 102 (56.98) | 280 (66.99) | 54 (70.13) | |
| Female | 77 (43.02) | 138 (33.01) | 23 (29.87) | |
| Hypertension | 70 (39.11) | 192 (45.93) | 35 (45.45) | 0.275** |
| Diabetes mellitus | | | | 0.108** |
| Type 2 Diabetes Mellitus | 63 (35.2) | 184 (44.02) | 29 (37.66) | |
| T2DM + Diabetic Ketoacidosis | – | 4 (0.96) | – | |
| Cardiovascular Disease | | | | 0.432** |
| Dyslipidemia | 1 (0.56) | 2 (0.48) | – | |
| Heart failure | 4 (2.23) | 21 (5.02) | 1 (1.3) | |
| Ischemia Heart Disease | 14 (7.82) | 43 (10.29) | 8 (10.39) | |
| Rheumatoid Heart Disease | 1 (0.56) | 4 (0.96) | 2 (2.6) | |
| Cerebro Vascular Accident | 12 (6.7) | 20 (4.78) | 1 (1.3) | 0.183** |
| Renal disease | | | | 0.13** |
| Acute Kidney Injury | 1 (0.56) | 3 (0.72) | 3 (3.9) | |
| Chronic Kidney Disease | 21 (11.73) | 54 (12.92) | 9 (11.69) | |
| Liver disease | | | | 0.163** |
| Chronic Liver Disease | 3 (1.68) | −7 (1.67) | 1 (1.3) | |
| Decompensated Chronic Liver Disease | 3 (1.68) | – | 1 (1.3) | |
| DCLD with Pulmonary Hypertension | 1 (0.56) | 8 (1.91) | – | |
| Hepatitis | 5 (2.79) | 9 (2.15) | – | |
| Respiratory disease | | | | **0.003**** |
| Bronchial Asthma | 5 (2.79) | 7 (1.67) | 3 (3.9) | |
| Chronic Obstructive | 13 (7.26) | 19 (4.55) | 3 (3.9) | |
| Pulmonry Disease | | | | |
| Interstitial Lung Disease | – | – | 2 (2.6) | |
| Malignancy | | | | 0.435** |
| Any metastatic carcinoma | 1 (0.56) | 5 (1.2) | 2 (2.6) | |
| Any solid tumor | 11 (6.15) | 21 (5.02) | – | |
| Leukemia | 3 (1.68) | 9 (2.15) | – | |
| Lymphoma | 2 (1.12) | 5 (1.2) | – | |
| Myeloma | 1 (0.56) | 1 (0.24) | – | |
| Hypothyroidism | 14 (7.82) | 28 (6.7) | 7 (9.09) | 0.727** |
| History of transplantation | 1 (0.56) | 5 (1.2) | 2 (2.6) | 0.386** |
| Past surgical history | 46 (25.7) | 115 (27.51) | 26 (33.77) | 0.414** |
| Pallor | 63 (35.2) | 108 (25.84) | 16 (20.78) | **0.023**** |
| Icterus | 19 (10.61) | 30 (7.18) | 4 (5.19) | 0.239** |

*(Continued)*

**Table 2.** (Continued)

| Variables | Low dose (n = 179) | Usual dose (n = 418) | High dose (n = 77) | p value |
|---|---|---|---|---|
| Cyanosis | 1 (0.56) | 2 (0.48) | – | 0.815** |
| Clubbing | 2 (1.12) | 5 (1.2) | – | 0.629** |
| Lymphadenopathy | 3 (1.68) | 3 (0.72) | – | 0.354** |
| Edema | 38 (21.23) | 80 (19.14) | 14 (18.18) | 0.804** |
| Glascow Coma Scale | 15 (3-15) | 15 (3-15) | 15 (3-15) | **0.001*** |
| Dialysis requirement during hospitalization | 65 (36.31) | 176 (42.11) | 31 (40.26) | 0.417** |
| Number of dialysis | 0 (0-28) | 0 (0-39) | 0 (0-43) | 0.278* |
| Type of dialysis | | | | |
| CRRT | 1 (0.56) | 7 (1.6) | −1 (1.3) | 0.831** |
| Hemodialysis | 9 (5.03) | 20 (4.78) | – | |
| Sustained Low Efficiency Dialysis | 19 (10.61) | 61 (14.59) | 13 (16.88) | |
| SLED + HD | 36 (20.11) | 88 (21.05) | 17 (22.08) | |
| Duration of dialysis | | | | |
| 1.5–3 hrs | 2 (1.12) | 12 (2.87) | 1 (1.3) | 0.245** |
| 3–4 hrs | 7 (3.91) | 5 (1.2) | 1 (1.3) | |
| 3–5 hrs | 14 (7.82) | 33 (7.89) | 5 (6.49) | |
| 3–6 hrs | 25 (13.97) | 58 (13.88) | 15 (19.48) | |
| 5–6 hrs | 16 (8.94) | 61 (14.59) | 9 (11.69) | |
| 24 hrs | 1 (0.56) | 7 (1.67) | – | |
| Transferred from outside hospital | 52 (29.05) | 140 (33.49) | 21 (27.27) | 0.365** |
| History of recent admission | 20 (11.17) | 54 (12.92) | 10 (12.99) | 0.821** |
| Charlsons comorbidity index | 0 (0-10) | 3 (0-11) | 2 (0-10) | 0.108* |
| APACHE II score | 14 (1-38) | 14 (0-130) | 14 (2-95) | 0.221* |
| SOFA score | 5 (0-14) | 5 (0-18) | 6 (0-15) | **0.056*** |
| Resistance pattern | | | | |
| Susceptible | 31 (17.32) | 66 (15.79) | 9 (11.69) | 0.362** |
| Multi Drug Resistance | 13 (7.26) | 22 (5.26) | 8 (10.39) | |
| Extended Drug Resistance | 128 (71.51) | 311 (74.4) | 59 (76.62) | |
| Pathogens | | | | |
| CRAB | 52 (29.37) | 189 (45.1) | 48 (62.33) | 0.278** |
| CRE | 28 (15.81) | 114 (27.2) | 20 (25.97) | |
| CRPA | 25 (14.12) | 107 (25.53) | 18 (23.37) | |
| Combination therapy with polymyxin B | | | | |
| Tigecycline | 14 (7.9) | 37 (8.83) | 10 (12.98) | 0.124** |
| Cefta-Avi+Aztreonam | 12 (6.7) | 25 (5.96) | 8 (10.38) | |
| Meropenem | 8 (4.5) | 18 (4.29) | 12 (15.58) | |
| Other β lactams | 16 (9.03) | 24 (5.72) | 4 (5.19) | |

APACHE II: Acute physiological and chronic health evaluation; SOFA: sequential organ failure assessment; CRAB: carbapenem resistant Acinetobacter baumannii; CRE: carbapenem resistant Enterobacteriaceae; CRPA: carbapenem resistant pseudomonas aeruginosa, *Kruskal-wallis test, **Chi-square test.

**Table 3. Clinical outcomes in the patients after polymyxin B therapy (pre-matching).**

| Variables | Low dose (n=179) | Usual dose (n=418) | High dose (n=77) | p value |
|---|---|---|---|---|
| Outcome of therapy | | | | |
| Death | 86 (48.04) | 255 (61) | 48 (62.34) | **0.009**** |
| Improved | 93 (51.96) | 163 (39) | 29 (37.66) | |
| Length of hospital stay | 25 (6-126) | 20 (3-111) | 21 (3-103) | **0.004*** |
| Septic shock after polymyxin B therapy | 70 (39.11) | 204 (48.8) | 37 (48.05) | 0.299** |
| AKI due to sepsis | 61 (34.08) | 154 (36.84) | 34 (44.16) | 0.273** |
| Microbiological clearance | 21 (11.73) | 54 (12.92) | 12 (15.58) | 0.325** |
| Susceptibility to empirical therapy | 93 (51.96) | 241 (57.66) | 33 (42.86) | **0.049**** |
| Vasopressor free days after poly B therapy | 3 (0-46) | 4 (0-50) | 3 (0-24) | 0.201* |
| Ventilator free days after poly B therapy | 9 (0-106) | 3 (0-76) | 4 (0-56) | **0.00003*** |
| ICU free days after poly B therapy | 9 (0-111) | 4 (0-77) | 6 (0-56) | **0.012*** |
| Duration of ECMO | 0 (0) | 0 (0-35) | 0 (0-3) | 0.403* |
| Duration of poly B therapy | 7 (1-24) | 7 (1-27) | 8 (2-30) | 0.886* |
| Other complications due to sepsis after Polymyxin B therapy | | | | |
| ARDS | 18 (0.06) | 54 (12.92) | 14 (18.18) | |
| DIC | 3 (1.68) | 1 (0.24) | 2 (2.6) | |
| MODS | 15 (8.38) | 36 (8.61) | 13 (16.88) | |
| Cardiac complications | 11 (6.15) | 7 (1.67) | – | |
| Severe encephalopathy | 12 (6.7) | 23 (5.5) | 5 (6.49) | |
| Thromboembolism | 1 (0.56) | 4 (0.96) | – | |
| Multiple complications | 6 (3.35) | 40 (9.57) | 10 (12.99) | |
| Pneumothorax | – | 1 (0.24) | – | |

AKI: acute kidney injury; ECMO: extracorporeal membrane oxygenation; ARDS: acute respiratory distress syndrome; DIC: disseminated intravascular coagulation; MODS: multi-organ dysfunction syndrome, *Kruskal-wallis test, **Chi-square test.

### 3.4. Interpretation of end-point outcomes in dialysis-requiring patients

In the matched cohort of dialysis-requiring patients, comparative analysis clinical outcomes across different polymyxin B dosing strategies did not show statistically significant differences in 28-day mortality. However, microbiological clearance was significantly higher in the low-dose group compared to the usual-dose group (OR=1.27; 95%CI=[1.06–2.21];p=0.006), suggesting improved eradication of pathogens with low dose. No statistically significant difference was observed in microbiological clearance between the other two pairwise comparison groups. Patients receiving low dose regimen had significantly higher ventilator-free days than those receiving usual dose [8.6(0–106) vs 2(0–76);p=0.002]. Similarly, ICU-free days were significantly higher in the low-dose group compared to the usual-dose group [9.8(0–111) vs 3.5(0–77);p=0.047]. Vasopressor-free days showed a marginally significant difference between the low and usual dose groups [3(0–46) vs 3(0–50);p=0.05]. No significant differences in vasopressor-free days, ventilator-free days and ICU free days were noted in the other two pairwise groups. S4 Table gives the end point outcomes after polymyxin B therapy in the dialysis requiring cohort patients.

Kaplan-Meier curves did not show any statistically significant differences in 28-day survival between all the three pairwise comparison groups. S2 Fig. gives the Kaplan-Meier curves of all the comparison groups. We also plotted a Kaplan-Meier survival curves stratified across dialysis modalities (S3 Fig). Among patients undergoing IHD, low-dose strategy was associated with a trend toward improved survival compared to the usual dose. In contrast, within the SLED group, high-dose therapy showed poorest survival, with the majority of deaths occurring within the first few

**Table 4. Clinical outcomes in the patients after polymyxin B therapy (post-matching).**

| Variables | Usual Vs Low dose (n = 358) | | p value | High Vs Low dose (n = 154) | | p value | High Vs Usual dose (n = 154) | | p value |
|---|---|---|---|---|---|---|---|---|---|
| | Usual dose (n = 179) | Low dose (n = 179) | | High dose (n = 77) | Low dose (n = 77) | | High dose (n = 77) | Usual dose (n = 77) | |
| Outcome of therapy | | | 0.0042 | | | 0.0023 | | | |
| Death | 109 (60.89) | 86 (48.04) | | 48 (62.34) | 37 (48) | | 48 (62.34) | 47 (61) | 0.0031 |
| Improved | 70 (39.1) | 93 (51.96) | | 29 (37.66) | 40 (52) | | 29 (37.66) | 30 (39) | |
| Length of hospital stay | 20 (5-95) | 25 (6-126) | 0.004 | 21 (3-103) | 25 (6-100) | 0.012 | 21 (3-103) | 17 (3-94) | 0.025 |
| Septic shock after polymyxin B therapy | 87 (48.6) | 70 (39.11) | 0.214 | 37 (48.05) | 30 (39) | 0.314 | 37 (48.05) | 38 (49.35) | 0.03 |
| AKI due to sepsis | 66 (36.87) | 61 (34.08) | 0.324 | 34 (44.16) | 26 (33.7) | 0.251 | 34 (44.16) | 28 (36.36) | 0.241 |
| Microbiological clearance | 23 (12.84) | 21 (11.73) | 0.214 | 12 (15.58) | 9 (11.68) | 0.351 | 12 (15.58) | 10 (13) | 0.365 |
| Susceptibility to empirical therapy | 103 (57.5) | 93 (51.96) | 0.132 | 33 (42.86) | 40 (52) | 0.142 | 33 (42.86) | 44 (57.14) | 0.265 |
| Vasopressor free days after poly B therapy | 4 (0-45) | 3 (0-46) | 0.213 | 3 (0-24) | 3 (0-40) | 0.215 | 3 (0-24) | 4 (0-45) | 0.142 |
| Ventilator free days after poly B therapy | 4 (0-70) | 9 (0-106) | 0.0005 | 4 (0-56) | 8 (0-95) | 0.003 | 4 (0-56) | 3 (0-70) | 0.004 |
| ICU free days after poly B therapy | 5 (0-75) | 9 (0-111) | 0.025 | 6 (0-56) | 7 (0-90) | 0.12 | 6 (0-56) | 4 (0-70) | 0.03 |
| Duration of ECMO | 0 (0-35) | 0 (0) | 0.321 | 0 (0-3) | 0 (0) | 0.421 | 0 (0-3) | 0 (0-35) | 0.214 |
| Duration of poly B therapy | 7 (1-27) | 7 (1-24) | 0.754 | 8 (2-30) | 7 (1-23) | 0.741 | 8 (2-30) | 7 (1-25) | 0.354 |
| Other complications due to sepsis after Polymyxin B therapy | | | | | | | | | |
| ARDS | 23 (12.84) | 18 (0.06) | | 14 (18.18) | 8 (10.38) | | 14 (18.18) | 10 (13) | |
| DIC | 1 (0.55) | 3 (1.68) | | 2 (2.6) | 1 (1.3) | | 2 (2.6) | – | |
| MODS | 6 (3.35) | 15 (8.38) | | 13 (16.55) | 6 (7.8) | | 13 (16.55) | 7 (9.1) | |
| Cardiac complications | 3 (1.67) | 11 (6.15) | | – | 5 (6.5) | | – | 1 (1.3) | |
| Severe encephalopathy | 10 (5.58) | 12 (6.7) | | 5 (6.49) | 5 (6.5) | | 5 (6.49) | 4 (5.19) | |
| Thromboembolism | 2 (1.11) | 1 (0.56) | | – | – | | – | 1 (1.3) | |
| Multiple complications | 17 (9.49) | 6 (3.35) | | 10 (12.99) | 3 (3.9) | | 10 (12.99) | 7 (9.1) | |
| Pneumothorax | – | – | | – | – | | – | – | |

AKI: acute kidney injury; ECMO: extracorporeal membrane oxygenation; ARDS: acute respiratory distress syndrome; DIC: disseminated intravascular coagulation; MODS: multi-organ dysfunction syndrome.

**Table 5. End point outcomes after polymyxin B therapy in all the included cohort patients (after propensity score matching, n = 666).**

| Clinical outcomes | Usual Vs Low dose (n = 358) | | High Vs Low dose (n = 154) | | High Vs Usual dose (n = 154) | |
|---|---|---|---|---|---|---|
| | | p value | | p value | | p value |
| 28-day mortality [Cox proportional hazard (95% CI)] | 1.47 (1.11-1.95) | 0.007 | 1.25 (0.81-1.92) | 0.314 | 1.01 (0.67-1.53) | 0.944 |
| Microbiological clearance [Odds ratio (95% CI)] | 1.17 (0.63-2.21) | 0.617 | 1.24 (0.50-3.12) | 0.646 | 1 (0.42-2.41) | 1.00 |
| Ventilator free days (Median IQR) | 9 (0–106) vs 3 (0–76) | 0.00002 | 9 (0–106) vs 4 (0–56) | 0.25 | 3 (0–76) vs 4 (0–56) | 0.74 |
| ICU free days (Median IQR) | 9 (0–111) vs 4 (0–77) | 0.003 | 9 (0–111) vs 6 (0–56) | 0.433 | 4 (0–77) vs 6 (0–56) | 0.887 |
| Vasopressor free days (Median IQR) | 3 (0–46) vs 4 (0–50) | 0.005 | 3 (0–46) vs 3 (0–24) | 0.457 | 4 (0–50) vs 3 (0–24) | 0.800 |

weeks. In patients who were on both the modes of dialysis, high-dose regimens consistently exhibited the lowest cumulative survival, while low-dose group maintained a relative advantage. Though the curves inferred these results, log-rank test did not show any statistical significance. These findings suggest that aggressive dosing in dialysis patients may not translate into improved survival.

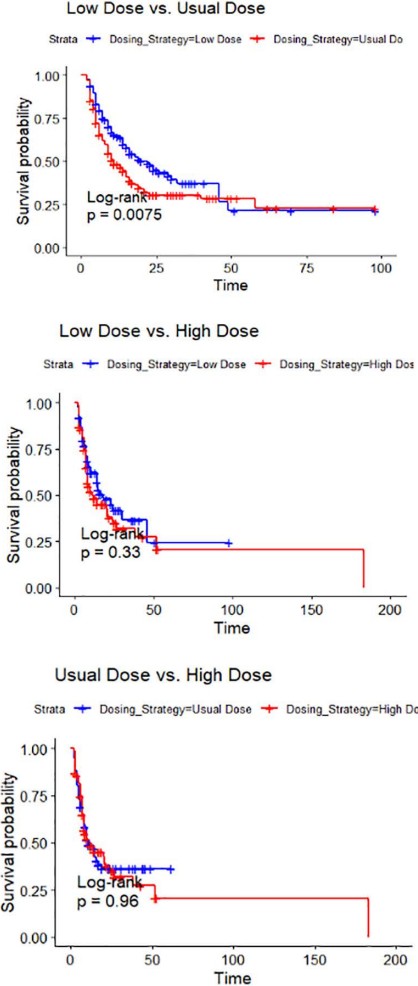

Survival probabilities were compared using the log-rank test; hazard ratios were derived from Cox proportional-hazards regression adjusted for matched covariates. HR-hazard ratio; CI: confidence interval; LU-lakh units

**Fig 2. Kaplan Meier analysis of 28-day mortality between patients comparing low, usual and high dosing strategy of polymyxin B after matching. (A)** Low dose Vs. Usual dose **(B)** Low dose Vs High dose **(C)** Usual dose Vs High dose.

### 3.5. Subgroup and sensitivity analysis for the primary endpoint

Sensitivity analyses using pairwise 1:2 and 1:3 nearest neighbouring propensity score matching approaches were conducted to confirm the robustness of the primary findings. In the 1:2 matching analysis, there was a significant increased risk of 28-day mortality in the usual (HR = 1.48; 95% CI:[1.07–1.85];p = 0.006) and high-dose groups (HR = 1.45; 95% CI: [1.06–1.87];p = 0.05) when compared to the low dose group. There was no significant difference in mortality between the usual and high-dose groups. In the 1:3 matching analysis, the association between usual and low dose remained consistent (HR = 1.48; 95%CI: [1.12–1.96];p = 0.006), when compared to the high and low dose groups. However, the comparison between the other two pairwise comparison groups did not show any statistical significance. These findings reinforce that low-dose polymyxin B may be associated with decreased risk of mortality compared to the usual-dose regimens, while differences between usual and high dose remains inconclusive. S5 and S6 Tables give the sensitivity analysis for 28-day mortality for both 1:2 and 1:3 nearest neighbour matching.

Gender-based subgroup analysis showed a consistent trend towards increased 28-day mortality in both males (HR = 1.56; 95%CI:[1.08–2.27];p = 0.018) and females (HR = 1.59; 95% CI:[1.01–2.48];p = 0.044) receiving usual-dose polymyxin B compared to the low dose. No significant differences were observed in the other pairwise comparisons for both the gender. These findings indicate that the association between low dose polymyxin B and decreased mortality persists across gender subgroups. S7 Table gives the subgroup analysis for 28-day mortality categorized based on gender.

## 4. Discussion

In this retrospective cohort study involving critically ill patients with gram-negative sepsis, we investigated the comparative effectiveness of three dosing strategies of polymyxin B- low, usual and high doses, on clinical and microbiological outcomes. Our findings suggest that the low-dose polymyxin B regimen was associated with significantly lower 28-day mortality and more favourable secondary outcomes such as vasopressor, ventilator and ICU-free days, when compared to the other dosing groups. These results remained consistent in sensitivity analyses, underscoring the robustness of our primary findings. Also, rather than focusing on post-hoc power, we emphasize the observed effect sizes and their corresponding 95% confidence interval as measures of both statistical precision and clinical relevance.

Although current international guidelines recommend a weight-based maintenance dose of 1.25–1.5 mg/kg every 12 hours for polymyxin B, there is a growing concern about achieving therapeutic targets without increasing toxicity [7]. The guideline-based recommendations largely stem from pharmacokinetic simulations rather than clinical outcome data, and many real-world studies, including ours, have reported considerable variability in drug exposures and outcomes even within these recommended dosing ranges [9,20]. Recent pharmacokinetic and TDM-focused studies emphasize the large interpatient variability in polymyxin B exposure and the potential utility of TDM or PK-guided dosing to optimise therapeutic windows. Clinical TDM studies and population-PK analyses have shown that model-based dose adjustment improves target attainment and highlight the limitations of fixed-dose approaches, particularly in special populations such as those receiving extracorporeal support or renal replacement therapy [21,22].

Our results align with those of Lie et al., who observed no statistically significant difference in 28-day mortality between high and low-dose polymyxin B groups, although prolonged survival was slightly better in high-dose recipients over 180 days [10]. Several additional studies similarly explore mortality outcomes in relation to polymyxin B. Rigatto et al., [23] reported that daily doses ≥150 mg/day were associated with increased AKI but not definitively with higher mortality. In patients with bloodstream infections due to carbapenem-resistant gram-negative rods, doses <1.3 mg/kg/day were associated with greater 30-day mortality compared to higher doses [24]. Additionally, a high-dose cohort study using ~30,000 IU/kg/day polymyxin B found promising survival but at the cost of significant AKI incidence [25]. In contrast, our findings indicate that low-dose regimens were associated with comparable clinical outcomes to the usual dose in terms of 28-day mortality, given the variations in patient selection, severity of illness, or local resistance patterns. Cai et al., reported microbiological eradication in 95.8% of bloodstream infections treated with high dose polymyxin B [24]. Microbiological clearance rates in our study did not significantly differ across groups, suggesting that lower doses may be sufficient to achieve bacteriological efficacy in selected populations, especially when combination therapy is employed.

The finding of higher 28-day mortality among patients receiving the usual-dose polymyxin B regimen should be interpreted with caution. Confounding by indication is the most plausible explanation for this association. In routine clinical practice, clinicians may escalate polymyxin B dosing in patients perceived to have more severe illness, poor early response, or unfavourable prognostic features that are not fully captured by severity scores such as APACHE II or SOFA. Although propensity score matching was used to balance measured confounders, residual confounding from unmeasured clinical factors likely remains. Therefore, the observed association should not be interpreted as evidence of a harmful effect of usual-dose therapy but rather as a reflection of treatment selection bias inherent to retrospective observational studies.

Among dialysis-requiring patients, the low-dose strategy demonstrated favourable trends in ventilator-and ICU-free days, as well as a higher microbiological clearance rate compared to the usual dose. However, 28-day mortality differences were not statistically significant in this subgroup. This observation is critical, as optimal polymyxin B dosing in patients undergoing renal replacement therapy remains an unresolved clinical question. The lack of consensus on renal dose adjustment, despite evidence suggesting limited renal elimination of polymyxin B, often results in empirically reduced dosing in clinical practice [7]. Our findings argue on supplemental dosing in dialysis-dependent patients, considering the absence of survival benefit and the potential risk of toxicity. We observed all cases of AKI reported in the medical records following the initiation of polymyxin B therapy. However, due to the retrospective design of the study, it was not possible to determine whether the reported AKI was attributable to the disease itself or to the drug. Furthermore, a causal relationship between AKI and polymyxin B administration could not be established, as no dechallenge or rechallenge was performed. This limitation is particularly important given the critical condition of the study population, in which multiple concurrent factors could have contributed to renal dysfunction.

The observation that patients receiving the usual-dose regimen experienced higher mortality compared with those on lower doses was unexpected. This finding likely reflects a combination of clinical and pharmacological factors rather than a true dose-response effect. Confounding by indication is the most plausible explanation, as clinicians may have preferentially prescribed higher doses to patients perceived as more severely ill or with poor prognostic indicators not fully captured by APACHE II or SOFA scores. Additionally, the potential for dose-related adverse effects, particularly nephrotoxicity in patients with renal impairment, cannot be excluded. Local fixed dose prescribing practices may also have resulted in inadvertent overdosing in smaller patients, amplifying toxicity risk. Collectively, these factors could explain the paradoxical association and underscore the need for randomized controlled trials to delineate true dose-outcome relationships.

These findings also contribute to the ongoing debate on the exposure-response relationship of polymyxin B. Pharmacokinetic and pharmacodynamic studies have consistently demonstrated high interindividual variability and suboptimal target attainment even with guideline-recommended doses [9]. Our results add a clinical perspective to these observations, emphasizing the importance to individualized dosing, particularly in patients with renal impairment or multi-organ dysfunction who are prone to drug accumulation. Although lower doses were not associated with worse outcomes in our cohort, the absence of systematic toxicity data precludes conclusions about safety benefits. Thus, the rationale for exploring lower-dose regimens remains hypothesis-generating and warrants confirmation in prospective studies that assess both efficacy and safety outcomes.

As a single-centre study, our results reflect local prescribing habits, patient mix and dialysis practices, which may limit generalizability. Nevertheless, they offer valuable real-world insight into polymyxin B use in critically ill patients with renal impairment in a high burden setting and the urgent need for multicentre randomized trials to establish optimal, safe and effectives dosing strategies.

### 4.1. Limitations

This was a retrospective single-centre study, which may limit the generalizability of our findings, as local prescribing practices, pathogen distribution, and dialysis protocols may vary across institutions. However, the methodological approach employed to evaluate dosing strategies can be adapted and applied in future studies conducted in different settings. Although propensity score matching was performed to minimize bias from measured variables, residual and unmeasured confounding cannot be excluded. Physician discretion in dose selection, based on clinical impressions not captured by APACHE II or SOFA scores, may have influenced outcomes. Additionally, microbiological parameters such as MIC values for polymyxin B and data on concurrent nephrotoxic agents (e.g., aminoglycosides, vancomycin, loop diuretics) were not consistently available and could not be incorporated into the model. Both factors could plausibly influence treatment selection and patient outcomes; therefore, the observed associations should be interpreted as hypothesis-generating rather than casual.

An important limitation of this study is the absence of body weight for a substantial proportion of patients, which prevented normalization of polymyxin B dosing on a mg/kg basis. As a result, the fixed-dose categories used in this analysis represent centre-specific prescribing constructs rather than pharmacologically standardized dosing thresholds. This limitation restricts the interpretability of dose-response relationships and limits the generalizability of our findings to settings where weight-based dosing is routinely implemented. Accordingly, the observed associations should be interpreted cautiously and viewed as hypothesis-generating. Furthermore, pharmacokinetic data to correlate administered doses with serum drug concentrations or toxicity endpoints were lacking, and systematic assessment of nephrotoxicity and neurotoxicity could not be performed. AKI events, though documented in medical records, could not be reliably attributed to polymyxin B exposure versus sepsis-related organ dysfunction or hemodynamic instability. This precludes any conclusions regarding the comparative safety or nephrotoxicity advantage of lower versus higher polymyxin B dosing regimens and our findings should not be interpreted as evidence of reduced toxicity. Although, missing data were handled by excluding variables with >20% missingness and imputing medians (continuous variables) or modes (categorical variables) when <20% was missing, this approach may still introduce bias and should be acknowledged as a limitation.

Microbiological data such as infection source, MIC values and timing of follow-up cultures were not uniformly documented, which may have introduced classification bias when assessing clearance. The use of fixed-dose categories, while reflective of local practice, deviates from guideline-recommended weight-adjusted dosing and underscores the need for future studies with complete dosing, body weight, and pharmacokinetic data to define true exposure-response relationships. Finally, the dialysis subgroup was heterogenous, comprising patients receiving SLED, CRRT and IHD; which prevented in performing further analysis after matching; differences in drug clearance between modalities may have contributed to residual confounding that could not be fully accounted for in our analysis.

### 4.2. Strengths and future implications

Our study has several strengths, including robust propensity score matching to minimize confounding and extensive sensitivity and subgroup analyses to confirm the consistency of findings. It is one of the large real-world datasets comparing three dosing tiers of polymyxin B in a critically ill population, including a substantial proportion of patients undergoing dialysis. Also, our study reflects real-world prescribing practices and patient heterogeneity across dosing regimens, thereby increasing the external validity of our findings and making them more applicable to resource-limited settings. Future prospective trials incorporating therapeutic drug monitoring, especially in special populations such as those on renal replacement therapy, are warranted. Also, they could incorporate standardized toxicity endpoints to better delineate the benefit-risk profile of different dosing strategies. Additionally, efforts to define the minimal effective dose that achieves optimal efficacy while minimizing toxicity should be prioritized. Given the global increase in antimicrobial resistance and limited therapeutic options, refining polymyxin B dosing strategies remains a crucial are of clinical research.

### 5. Conclusion

In this single-centre retrospective study, though lower fixed doses of polymyxin B were associated with favourable clinical outcomes, the absence of standardized toxicity assessment and the observational study design preclude conclusions regarding safety or therapeutic equivalence. These results should be interpreted as hypothesis-generating rather than practice-changing. Our findings support the feasibility and importance of investigating lower dose polymyxin B regimens in future RCTs.

---

Key learning points

**What was known**

Polymyxin B is a last-line antibiotic for multidrug resistant gram-negative infections, but optimal dosing in patients with renal impairment—particularly those on dialysis—remains uncertain due to limited clinical outcome data.

---

                                                                                

**This study adds**

This propensity score-matched cohort study suggests that low-dose polymyxin B was associated with comparable clinical outcomes compared to conventional or high-dose regimens. In terms of microbiological clearance, low-dose group did not show a statistically significant difference compared to the other groups; even in dialysis-dependent patients, with safety outcomes not systematically evaluated.

**Potential impact**

This study highlights the feasibility of lower fixed dose polymyxin B regimens in critically ill patients with renal impairment, suggesting they may achieve comparable outcomes with no systematic data on toxicity. These findings provide a basis for future randomized trials and support evidence-based antibiotic stewardship in ICUs.

## Supporting information

**S1 Table. Missing number for included variables.**
(DOCX)

**S2 Table. Covariate balance after propensity score matching assessed using standardized mean differences (SMDs).**
(DOCX)

**S3 Table. Baseline study characteristics of the patients between low and usual group (post-matching).**
(DOCX)

**S4 Table. End point outcomes after polymyxin B therapy in the dialysis requiring cohort patients (after propensity score matching, n = 254).**
(DOCX)

**S5 Table. Sensitivity analysis for 28-day mortality for all the included cohort patients after polymyxin B therapy (after propensity score matching using pairwise 1:2 nearest neighbour matching).**
(DOCX)

**S6 Table. Sensitivity analysis for 28-day mortality for all the included cohort patients after polymyxin B therapy (after propensity score matching using pairwise 1:3 nearest neighbour matching).**
(DOCX)

**S7 Table. Subgroup analysis for 28-day mortality for all the included cohort patients after polymyxin B therapy categorized based on gender (after propensity score matching).**
(DOCX)

**S1 Fig. Culture reports pre- and post-polymyxin B therapy with various dosing strategies among all the included cohort patients (i) Blood (ii) endotracheal tube (iii) urine (iv) wound swab (v) broncho-alveolar lavage (vi) Tissue (vii) body fluids (viii) catheter tip (ix) pus (x) sputum (xi) nasal swab.**
(DOCX)

**S2 Fig. Kaplan Meier analysis of 28-mortality status, 28-day mortality between patients requiring dialysis comparing low, usual and high dosing strategy of polymyxin B after matching.** (A) Low dose Vs Usual dose (B) Low dose Vs High dose (C) Usual dose Vs High dose.
(DOCX)

**S3 Fig. Kaplan Meier analysis of 28-mortality status, 28-day mortality between patients with various dosing strategies requiring different types of dialysis, comparing low, usual and high dosing strategy of polymyxin B after matching.** (A) Hemodialysis (B) Sustained low efficiency dialysis (C) Hemodialysis + Sustained low efficiency dialysis. (DOCX)

## Acknowledgments

The authors are thankful to Manipal College of Pharmaceutical Sciences, Manipal Academy of Higher Education (MAHE), Manipal, Department of Nephrology and Department of Critical Care Medicine, Kasturba Medical College, MAHE, Manipal; Medical Records Department, Kasturba Hospital, MAHE, Manipal.

Patient consent: This study does not include consent from patients owing to its study design.

## Author contributions

**Conceptualization:** Vishal Shanbhag.

**Data curation:** Asha K Rajan.

**Formal analysis:** Asha K Rajan, Beven Nelson.

**Investigation:** Asha K Rajan, Vishal Shanbhag, Vijayanarayana Kunhikatta, Girish Thunga.

**Methodology:** Asha K Rajan, Vishal Shanbhag, Ravindra Prabhu Attur, Varun Kumar SG.

**Project administration:** Varun Kumar SG, Girish Thunga.

**Resources:** Vishal Shanbhag, Ravindra Prabhu Attur.

**Software:** Asha K Rajan, Vijayanarayana Kunhikatta, Beven Nelson, Varun Kumar SG.

**Supervision:** Vishal Shanbhag, Vijayanarayana Kunhikatta, Ravindra Prabhu Attur, Souvik Chaudhuri, Girish Thunga.

**Validation:** Asha K Rajan, Ravindra Prabhu Attur, Beven Nelson.

**Writing – original draft:** Asha K Rajan.

**Writing – review & editing:** Asha K Rajan, Vishal Shanbhag, Vijayanarayana Kunhikatta, Ravindra Prabhu Attur, Beven Nelson, Varun Kumar SG, Souvik Chaudhuri, Girish Thunga.

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
