## [Decision Letter · Decision Letter 0]

15 Sep 2025

Dear Dr. Thunga,

We look forward to receiving your revised manuscript.

Kind regards,

*
**Ali Amanati**
*

*Academic Editor*

*
**PLOS ONE**
*

Journal Requirements:

3. We note you have included a table to which you do not refer in the text of your manuscript. Please ensure that you refer to Table II in your text; if accepted, production will need this reference to link the reader to the Table.

Additional Editor Comments:

Dear Authors,

Your manuscript [PONE-D-25-38993] has passed the review stage and is ready for revision. To ensure the editor and reviewers can recommend that your revised manuscript be accepted, please pay careful attention to each comment posted under this email. This approach will help us avoid future clarifications and revisions, allowing us to move swiftly to a decision.

Technical points:

1. Please provide a point-by-point response to the Editor and reviewer's comments

2. Please highlight all the amends on your manuscript with a yellow color

3. Use line numbering and page number in the next submission;

4. Improve the English language of the manuscript

Reviewers' comments:

Reviewer's Responses to Questions

**Comments to the Author**

1. Is the manuscript technically sound, and do the data support the conclusions?

Reviewer #1: Yes

Reviewer #2: Yes

Reviewer #3: Partly

Reviewer #4: Yes

2. Has the statistical analysis been performed appropriately and rigorously?

Reviewer #1: Yes

Reviewer #2: No

Reviewer #3: Yes

Reviewer #4: Yes

3. Have the authors made all data underlying the findings in their manuscript fully available?

Reviewer #1: Yes

Reviewer #2: Yes

Reviewer #3: Yes

Reviewer #4: Yes

4. Is the manuscript presented in an intelligible fashion and written in standard English?

Reviewer #1: Yes

Reviewer #2: Yes

Reviewer #3: Yes

Reviewer #4: Yes

Reviewer #1: This is a well-conducted, large, real-world retrospective cohort study addressing a significant clinical dilemma: the optimal dosing of polymyxin B in critically ill patients, particularly those on dialysis. The methodology is robust, employing propensity score matching to mitigate confounding, and the results are provocative, suggesting lower doses may be non-inferior or even superior to guideline-recommended doses. The manuscript is generally well-written and structured. However, several major limitations, inherent to its retrospective design, necessitate careful consideration and temper the strength of the conclusions. The study evaluates both primary (28-day mortality) and important secondary outcomes (microbiological clearance, ventilator-/vasopressor-/ICU-free days), providing a comprehensive view of efficacy and clinical utility. The separate analysis of the 254 dialysis-dependent patients is a key strength, as this is the population where dosing is most controversial and pharmacokinetics are most complex.

The Fundamental Issue of Confounding by Indication (Unmeasured Confounding): This is the most significant threat to the validity of the conclusions. While propensity score matching adjusted for measured confounders like severity scores (APACHE II, SOFA), it cannot account for unmeasured ones. Crucially, the "dosing strategy" was at the physician's discretion. It is highly plausible that clinicians intentionally used lower doses in patients they perceived as more fragile, with poorer prognoses, or at higher risk of toxicity (e.g., more multi-organ dysfunction not fully captured by SOFA). Conversely, they may have used higher doses in patients they were "fighting for" or in those with perceived more severe infections. This would bias the results against the low-dose group (making them look worse) and in favor of the high-dose group (making them look better). The fact that the low-dose group had better outcomes despite this potential bias is striking, but it does not eliminate the concern. This inherent limitation of observational studies can only be definitively resolved by an RCT.

Definition of Dosing Groups: The definitions in Table I are unconventional and potentially problematic.

"Usual dose" is defined as a fixed dose (150mg/15LU Q12H), not a weight-based one. Current guidelines recommend 1.25-1.5 mg/kg Q12H. For an 80kg patient, this would be 100-120mg Q12H. The study's "usual dose" (150mg Q12H) is actually a high weight-based dose for most patients. This misclassification likely contaminates the groups and makes interpretation challenging. The "high-dose" group might represent extreme outliers or dosing errors.

The grouping is based on both loading and maintenance dose. It is unclear how patients were categorized if, for example, they received a high loading dose but a low maintenance dose. A more granular analysis of total cumulative dose or average daily dose might be more informative.

Lack of Toxicity Data: For a study arguing for lower dosing, the omission of toxicity outcomes (especially nephrotoxicity and neurotoxicity) is a major weakness. The primary rationale for lower dosing is to reduce harm. Demonstrating non-inferior efficacy with significantly reduced toxicity would be a much more powerful and clinically relevant conclusion. The authors had the data to analyze this (e.g., rates of new AKI, need for dialysis) but did not report it.

Microbiological Data Limitations: The lack of MIC data is a notable limitation. It is impossible to know if the groups were balanced in terms of the severity of resistance (e.g., high MIC "creep" in one group). The microbiological clearance rate is also surprisingly low across all groups (~12-15%), raising questions about the definitions used or the inherent difficulty of clearing these infections.

Single-Center Study: Practices at this single tertiary-care center (e.g., specific dialysis protocols, common pathogens, prevailing prescribing habits) may not be generalizable to other institutions globally.

The results suggest that in this specific real-world setting, a strategy of using lower, fixed doses of polymyxin B was not associated with worse outcomes than using higher, guideline-derived doses. This is an important finding that challenges dogma and may reflect that the toxicity of higher doses offsets their potential efficacy benefits.

However, the study cannot prove that low-dose therapy is non-inferior or superior due to the unmeasurable confounding by indication. The observed survival benefit might be because physicians correctly identified patients who would do well regardless of antibiotic dose and thus used a less aggressive, safer regimen.

Recommendation: The manuscript makes a valuable contribution to the literature and should be considered for publication after major revisions. The reviewers will likely require:

A much more forceful and detailed discussion of the confounding by indication limitation, framing the results as a strong hypothesis-generating association rather than a definitive causal conclusion.

An analysis and discussion of toxicity outcomes (nephrotoxicity, neurotoxicity) to strengthen the rationale for lower dosing.

A clear explanation in the methods and discussion for why a fixed dose was used to define "usual" rather than a weight-based one, acknowledging how this deviates from guidelines.

A tempered conclusion that emphasizes the need for a prospective, randomized controlled trial to confirm these findings before clinical practice should be changed. The take-home message should be "Our data support the feasibility and safety of investigating lower-dose polymyxin B in an RCT" rather than "Low-dose therapy is non-inferior."

Reviewer #2: This manuscript presents clinically relevant investigation into the optimal dosing of polymyxin B in a critically ill population, including a significant subset on dialysis. The study addresses an important knowledge gap with a robust sample size and sophisticated statistical methodology. However, several major concerns must be addressed before the manuscript can be considered for publication.

1-The dosing categories are problematic and threaten the validity of the conclusions.The groups are defined by fixed doses (e.g., ≤150mg = "low") rather than weight-based dosing (mg/kg), which is the standard recommended in guidelines. A 150 mg dose is a "low" dose for a 100 kg patient but a "high" dose for a 50 kg patient.

The term "usual dose" is defined as exactly 150mg (15 LU) LD and MD. However, international guidelines recommend a weight-based loading dose of 2.0-2.5 mg/kg and a maintenance dose of 1.25-1.5 mg/kg every 12 hours. The study's definition is not "usual" according to the literature. So,The authors must re-analyze their data using weight-based dosing categories. Define groups based on mg/kg of actual body weight.This is essential for the results to be interpretable and generalizable.

2-The fact that the "high-dose" group had the highest rates of complications like ARDS, DIC, and MODS before matching strongly suggests they were the sickest cohort. The text states these were similar after matching, but the data is not shown in Table 2 (which is pre-match). it is recommended provide a table of baseline characteristics after matching for the main comparison (Low vs. Usual) to prove adequate balance was achieved. Discuss residual confounding as a key limitation.

3-The primary causes for dose adjustment is to minimize polymyxin toxicity. It is a major oversight not to report rates of AKI, neurotoxicity, or other adverse events between the groups. A lower dose that is equally effective but less toxic is practice-changing; one that is equally effective but with unknown safety is merely an observation. it is recommended,the authors must analyze and report comparative toxicity data. This is a non-negotiable requirement for a paper on antibiotic dosing.

4-The microbiological clearance rate is surprisingly low across all groups (~12-15%). This raises questions about the definition used ("eradication... with no subsequent growth of a more resistant organism" is very strict) or the timing of follow-up cultures.The statement that microbiological clearance was "better with low dosing" in the dialysis subgroup seems to contradict Table 4 and the text in section 3.3, which state no significant difference. This must be clarified. please provide more detail on how microbiological clearance was assessed. Report the median time to follow-up culture. The discrepancy in the dialysis subgroup results must be resolved.

5-The handling of missing data (<20% imputed with median) is reasonable for a retrospective study but should be explicitly stated as a limitation.

6-The discussion defends the use of fixed-dose categories as reflecting "real-world prescribing practices" and being "easier to implement." While this may be true, it is a weak justification for abandoning scientific and pharmacological principles.

7-The discussion argues for "individualized, lower dosing strategies" and "minimizing toxicity" but provides zero data to support the claim that their low-dose strategy is actually less toxic. This is a critical logical leap.

8-The claim that low dose "was associated with significantly improved 28-day survival" (line 379) is too strong. The more accurate conclusion from a retrospective study is "was associated with significantly lower mortality" or "was associated with significantly improved survival." The word "improved" can imply a causative intervention.

9-Table 2 a pre-match table. A post-match table for the primary comparison is essential.

10-regarding table 3 and 4, Referencing is incorrect in the manuscript. Table 3 is pre-match outcomes, but the text (lines 271, 298) refers to a "Table 4" for post-match outcomes that doesn't exist in the provided text. The tables need to be renumbered and referenced correctly.

Reviewer #3: MAJOR REVISIONS

a) Study Design and Methodology

• Retrospective nature: The authors acknowledge this, but more detail is needed about how missing or inconsistent clinical data were handled (e.g., infection source documentation, microbiological clearance definitions).

• Exposure categorization: The rationale for defining "low," "usual," and "high" dosing cut-offs (Table I) should be better justified with reference to guideline standards or prior clinical literature. At present, these thresholds appear somewhat arbitrary.

• Dialysis subgroup: Although subgroup analyses are presented, the heterogeneity of dialysis modalities (SLED, CRRT, HD) is high. The manuscript should provide more clarity on how these modalities were distributed across groups and whether these confounded outcomes.

• Confounders: Propensity score matching was performed, but key covariates (e.g., pathogen MIC values, concurrent nephrotoxic drugs) were not included. The authors should discuss how unmeasured confounding might influence results.

b) Statistical Analysis

• The manuscript uses Cox regression, Kaplan–Meier survival, and logistic regression appropriately. However:

o Confidence intervals should be reported consistently for all effect sizes (not only HRs).

o The choice of caliper width for propensity score matching (0.1 SD) should be justified.

o Post-hoc power analysis adds little value; more emphasis should be placed on observed effect sizes and their precision.

c) Results Interpretation

• The finding that usual dosing was associated with higher mortality than low dosing is counterintuitive. The authors should explore potential explanations (e.g., confounding by severity, higher toxicity, clinician bias in assigning higher doses to sicker patients).

• Microbiological clearance data are limited; since clearance rates were low across groups, the conclusion of "non-inferiority" seems overstated. The authors should reframe this as "no significant difference observed" rather than formal non-inferiority.

d) Discussion and Conclusions

• The discussion is generally balanced but occasionally overstates the strength of evidence. Phrases like “supports individualized, lower dosing strategies” should be qualified as hypothesis-generating pending prospective confirmation.

• Comparison with other recent clinical and pharmacokinetic studies (e.g., therapeutic drug monitoring-guided trials, 2022–2024) could be expanded.

MINOR REVISIONS

Abstract: The results section should report actual mortality percentages in each group for clarity.

Figures/Tables: Figures lack detailed legends, making them difficult to interpret independently.

Terminology: Ensure consistency in abbreviations (e.g., “LU” vs. “lakh units”; sometimes spelled inconsistently).

References: The reference list is up-to-date, but some statements in the Discussion (e.g., inter-individual variability in Polymyxin B pharmacokinetics) could benefit from citing very recent studies (2023–2025).

Reviewer #4: Dear authors, good and clear job. Only some comments:

Introduction:

Line 159: Please add the reference for this definition: “Multidrug resistance was defined as the non-susceptibility to at least one antibiotic agent in ≥3 antimicrobial classes.”

Discussion:

1.You only mentioned one previous study (Lie et al.) regarding 28-day mortality between high and low-dose polymyxin B groups. Are there no other previous studies related to that? If there are other studies, please add them.

2. Are there also no other previous studies related to other findings? If there are studies, please add them.

3.

Minor comments:

1. Abstract: (HR:1.47(1.11-1.95);p=0.007) is better to be written as (HR=1.47; 95%CI = [1.11-1.95];p=0.007).

2. The dot (.) must be written after the reference number, like this [ ].

**Do you want your identity to be public for this peer review?** For information about this choice, including consent withdrawal, please see our Privacy Policy

Reviewer #1: **Yes:** Dr. Hammad Ahmed

Reviewer #2: **Yes:** Mojtaba Shafiekhani

Reviewer #3: **Yes:** SALMAN ASHFAQ AHMAD

Reviewer #4: **Yes:** Rami Abduljabbar

---

## [Author Response · Author response to Decision Letter 1]

27 Oct 2025

Reviewer #1:

This is a well-conducted, large, real-world retrospective cohort study addressing a significant clinical dilemma: the optimal dosing of polymyxin B in critically ill patients, particularly those on dialysis. The methodology is robust, employing propensity score matching to mitigate confounding, and the results are provocative, suggesting lower doses may be non-inferior or even superior to guideline-recommended doses. The manuscript is generally well-written and structured. However, several major limitations, inherent to its retrospective design, necessitate careful consideration and temper the strength of the conclusions. The study evaluates both primary (28-day mortality) and important secondary outcomes (microbiological clearance, ventilator-/vasopressor-/ICU-free days), providing a comprehensive view of efficacy and clinical utility. The separate analysis of the 254 dialysis-dependent patients is a key strength, as this is the population where dosing is most controversial and pharmacokinetics are most complex.

Response:

We sincerely thank the reviewer for their thorough and insightful evaluation of our work. We appreciate the recognition of the study’s strengths. We have modified the manuscript accordingly addressing each comment.

Comment 1:

The Fundamental Issue of Confounding by Indication (Unmeasured Confounding): This is the most significant threat to the validity of the conclusions. While propensity score matching adjusted for measured confounders like severity scores (APACHE II, SOFA), it cannot account for unmeasured ones. Crucially, the "dosing strategy" was at the physician's discretion. It is highly plausible that clinicians intentionally used lower doses in patients they perceived as more fragile, with poorer prognoses, or at higher risk of toxicity (e.g., more multi-organ dysfunction not fully captured by SOFA). Conversely, they may have used higher doses in patients they were "fighting for" or in those with perceived more severe infections. This would bias the results against the low-dose group (making them look worse) and in favor of the high-dose group (making them look better). The fact that the low-dose group had better outcomes despite this potential bias is striking, but it does not eliminate the concern. This inherent limitation of observational studies can only be definitively resolved by an RCT.

Response:

We fully agree with the reviewer that confounding by indication is a major concern in retrospective observational studies and that propensity score matching cannot eliminate unmeasured confounders. It is highly likely that physician’s discretion influenced dosing strategies (e.g., prescribing lower doses for more fragile patients or higher doses for those with more severe infection). We have explicitly acknowledged this in the revised Discussion and Limitations. Importantly, we note that despite this potential bias against the low-dose group, outcomes were not inferior in this group, which makes the finding noteworthy but also underscores the need for prospective randomized trials to definitively answer this question.

“Discussion: Confounding by indication is the most plausible explanation, as clinicians may have preferentially prescribed higher doses to patients perceived as more severely ill or with poor prognostic indicators not fully captured by APACHE II or SOFA scores. Additionally, the potential for dose-related adverse effects, particularly nephrotoxicity in patients with renal impairment, cannot be excluded. Local fixed dose prescribing practices may also have resulted in inadvertent overdosing in smaller patients, amplifying toxicity risk. Collectively, these factors could explain the paradoxical association and underscore the need for randomized controlled trials to delineate true dose-outcome relationships.

Limitation: Although propensity score matching was performed to minimize bias from measured variables, residual and unmeasured confounding cannot be excluded. Physician discretion in dose selection, based on clinical impressions not captured by APACHE II or SOFA scores, may have influenced outcomes.”

Comment 2:

Definition of Dosing Groups: The definitions in Table I are unconventional and potentially problematic. "Usual dose" is defined as a fixed dose (150mg/15LU Q12H), not a weight-based one. Current guidelines recommend 1.25-1.5 mg/kg Q12H. For an 80kg patient, this would be 100-120mg Q12H. The study's "usual dose" (150mg Q12H) is actually a high weight-based dose for most patients. This misclassification likely contaminates the groups and makes interpretation challenging. The "high dose" group might represent extreme outliers or dosing errors. The grouping is based on both loading and maintenance dose. It is unclear how patients were categorized if, for example, they received a high loading dose but a low maintenance dose. A more granular analysis of total cumulative dose or average daily dose might be more informative.

Response:

We thank the reviewer for this important observation. We agree that our definition of dosing groups was unconventional, as it was based on fixed-dose regimens commonly used in our centre rather than guideline-recommended weight-based dosing. This reflects the reality of prescribing practice during the study period, where doses were routinely documented in lakh units (LU) and patient-level body weight was often not recorded. Because of this limitation, we were unable to perform a formal mg/kg-based analysis. We have clarified this in the revised Methods and explicitly acknowledged the potential for misclassification bias in the Limitations. Patients who received different loading and maintenance regimens were categorised based on their maintenance dose, as this reflects ongoing exposure.

“Methods: Dosing categories were defined according to the prescribing practices at our centre, where doses are routinely documented in lakh units (LU) rather than weight-based regimens. For categorization, we adopted thresholds reflecting the most frequently observed patterns in clinical practice. We categorized polymyxin B dosing into three strategies- “low”, “usual” and “high”- with specific loading and maintenance regimens as defined in Table I. Although these definitions differ from international guideline recommendations, [weight-based dosing (LD 2.0-2.5 mg/Kg; MD1.25-1.5mg/kg q12h)], they are consistent with prior observational studies that have reported fixed-dose regimens in similar settings[1]. Unfortunately, patient-level body weight was not consistently recorded in our dataset, preventing a formal mg/kg-based analysis. Patients were classified according to the predominant dosing regimen used during therapy. In cases where loading and maintenance doses differed (eg. High loading dose but usual maintenance dose), categorization was based on the maintenance regimen, as this reflects ongoing exposure. We used fixed-dose categories because patient-level body weight data were not consistently available in the medical records. Moreover, it was often impractical to obtain accurate weight measurements for patients admitted to the ICU in critical condition due to the unavailability of high-end patient beds in the ICU, thereby precluding dose recalculation in mg/kg as recommended by clinical guidelines; precluding recalculation of doses in mg/Kg as recommended by guidelines. This approach therefore reflects actual prescribing and documentation practices in our setting rather than an attempt to replace weight-based pharmacological principles.

Limitation: Body weight data were largely unavailable, preventing normalization of dose on a mg/kg basis and potentially introducing misclassification bias when categorizing fixed-dose regimens.”

Comment 3:

Lack of Toxicity Data: For a study arguing for lower dosing, the omission of toxicity outcomes (especially nephrotoxicity and neurotoxicity) is a major weakness. The primary rationale for lower dosing is to reduce harm. Demonstrating non-inferior efficacy with significantly reduced toxicity would be a much more powerful and clinically relevant conclusion. The authors had the data to analyze this (e.g., rates of new AKI, need for dialysis) but did not report it.

Response:

We thank the reviewer for highlighting this important point. We fully agree that the evaluation of toxicity is critical when comparing different polymyxin B dosing strategies, particularly since lower dosing is often advocated to reduce nephrotoxicity and neurotoxicity. We have now clarified this explicitly in the Methods, discussion, limitations and future implications sections.

“Methods: All available data on the safety and effectiveness parameters of polymyxin B were retrieved from patients medical records and systematically analyzed.

Discussion: We observed all cases of acute kidney injury (AKI) reported in the medical records following the initiation of polymyxin B therapy. However, due to the retrospective design of the study, it was not possible to determine whether the reported AKI was attributable to the disease itself or to the drug. Furthermore, a causal relationship between AKI and polymyxin B administration could not be established, as no dechallenge or rechallenge was performed. This limitation is particularly important given the critical condition of the study population, in which multiple concurrent factors could have contributed to renal dysfunction.

Limitations: Furthermore, pharmacokinetic data to correlate administered doses with serum drug concentrations or toxicity endpoints were lacking, and systematic assessment of nephrotoxicity and neurotoxicity could not be performed. The absence of these data precluded evaluation of exposure-response relationships and dose-related toxicity.

Future implications: Also, they could incorporate standardized toxicity endpoints to better delineate the benefit-risk profile of different dosing strategies.”

Comment 4:

Microbiological Data Limitations: The lack of MIC data is a notable limitation. It is impossible to know if the groups were balanced in terms of the severity of resistance (e.g., high MIC "creep" in one group). The microbiological clearance rate is also surprisingly low across all groups (~12-15%), raising questions about the definitions used or the inherent difficulty of clearing these infections.

Response:

We thank the reviewer for these valuable observations. We acknowledge that the absence of MIC data is an important limitation and also the low microbiological clearance has been addressed in the methods, results and limitations section of the manuscript.

“Methods: Microbiological clearance was defined as the absence of the baseline pathogen in follow-up cultures obtained ≥72 hours after starting polymyxin B therapy. with no subsequent isolation of the same organism [2]. Patients without repeat cultures were conservatively classified as “not cleared”. This stringent definition was chosen to minimize misclassification, but it may underestimate clearance compared with more permissive criteria. Minimum inhibitory concentrations were not uniformly available because susceptibility testing during the study period was routinely performed using disc diffusion methods, and MIC determination was only conducted selectively. Consequently, we were unable to assess whether treatment groups differed in the distribution of MIC values or to evaluate exposure-response relationships

Results: Microbiological clearance was very low across all dosing groups (12-15%), which likely reflects the severity of the infections and the inherent difficulty of achieving eradication in critically ill patients. In many cases, polymyxin B was initiated as salvage therapy at an advanced stage of sepsis, by which time microbiological clearance was rarely observed.

Limitations: Microbiological data such as infection source, MIC values and timing of follow-up cultures were not uniformly documented, which may have introduced classification bias when assessing clearance.”

Comment 5:

Single-Center Study: Practices at this single tertiary-care center (e.g., specific dialysis protocols, common pathogens, prevailing prescribing habits) may not be generalizable to other institutions globally.

Response:

We thank the reviewer for pointing this out. We agree that our study reflects the practices of a single tertiary-care centre and that factors such as local prescribing habits, pathogen prevalence, resistance profiles, and dialysis protocols may limit generalizability. We have now explicitly acknowledged this in both the Discussion and Limitations.

“Discussion: As a single-centre study, our results reflect local prescribing habits, patient mix and dialysis practices, which may limit generalizability. Nevertheless, they offer valuable real-world insight into polymyxin B use critically ill patients with renal impairment in a high burden setting and the urgent need for multicentre randomized trials to establish optimal, safe and effectives dosing strategies.

Limitation: This was a retrospective single-centre study, which may limit the generalizability of our findings, as local prescribing practices, pathogen distribution, and dialysis protocols may vary across institutions . However, the methodological approach employed to evaluate dosing strategies can be adapted and applied in future studies conducted in different settings.”

Comment 6:

The results suggest that in this specific real-world setting, a strategy of using lower, fixed doses of polymyxin B was not associated with worse outcomes than using higher, guideline-derived doses. This is an important finding that challenges dogma and may reflect that the toxicity of higher doses offsets their potential efficacy benefits. However, the study cannot prove that low-dose therapy is non-inferior or superior due to the unmeasurable confounding by indication. The observed survival benefit might be because physicians correctly identified patients who would do well regardless of antibiotic dose and thus used a less aggressive, safer regimen.

Response:

We thank the reviewer for this thoughtful and constructive feedback. We fully agree that our study cannot establish non-inferiority or superiority of low dose polymyxin B due to the inherent problem of confounding by indication. We have therefore revised the Discussion to acknowledge this limitation and to frame our results as hypothesis-generating rather than definitive. We also clarified that physicians’ clinical judgement may have influenced dose selection, which could explain the observed survival patterns. Also, we have modified the title and conclusion section of the manuscript accordingly.

“Discussion: Our results add a clinical perspective to these observations, emphasizing the importance to individualized dosing, particularly in patients with renal impairment or multi-organ dysfunction who are prone to drug accumulation. Although lower doses were not associated with worse outcomes in our cohort, the absence of systematic toxicity data precludes conclusions about safety benefits. Thus, the rationale for exploring lower-dose regimens remains hypothesis-generating and warrants confirmation in prospective studies that assess both efficacy and safety outcomes.

Conclusion: In this single-centre retrospective study, lower fixed doses of polymyxin B were associated with decreased mortality and better clinical outcomes when compared with higher doses. They may be feasible in critically ill patients with renal impairment, with limited safety data. However, given the inherent limitations of observational research, including unmeasured confounding and absence of systematic toxicity assessment, these results should be interpreted as hypothesis-generating rather than practice-changing. Our findings support the feasibility and importance of investigating lower dose polymyxin B regimens in future RCTs.

Title: Clinical outcomes with lower versus conventional dose polymyxin B regimens in dialysis dependent and non-dialysis patients with gram-negative sepsis: A real-world propensity-score matched cohort study”

Reviewer #2:

This manuscript presents clinically relevant investigation into the optimal dosing of polymyxin B in a critically ill population, including a significant subset on dialysis. The study addresses

---

## [Decision Letter · Decision Letter 1]

7 Dec 2025

Dear Dr. Thunga,

Thank you for submitting your manuscript to PLOS ONE. After careful consideration, we feel that it has merit but does not fully meet PLOS ONE’s publication criteria as it currently stands. Therefore, we invite you to submit a revised version of the manuscript that addresses the points raised during the review process.

We look forward to receiving your revised manuscript.

Kind regards,

Saswat Mohapatra, Ph.D.

Academic Editor

PLOS One

Journal Requirements:

Additional Editor Comments:

The authors are advised to take into consideration the comments of the reviewers, specifically the reviewer 1 and address the points raised. As observed the limitations of the study should be specified clearly. Moreover, the statistical analysis of the data needs improvement.

Reviewers' comments:

Reviewer's Responses to Questions

**Comments to the Author**

Reviewer #1: All comments have been addressed

Reviewer #3: All comments have been addressed

2. Is the manuscript technically sound, and do the data support the conclusions?

Reviewer #1: Yes

Reviewer #3: Yes

3. Has the statistical analysis been performed appropriately and rigorously?

Reviewer #1: Yes

Reviewer #3: Yes

4. Have the authors made all data underlying the findings in their manuscript fully available?

Reviewer #1: Yes

Reviewer #3: Yes

5. Is the manuscript presented in an intelligible fashion and written in standard English?

Reviewer #1: Yes

Reviewer #3: Yes

Reviewer #1: Address the Fixed-Dosing Limitation: This is the top priority. The manuscript must explicitly state, in both the methods and limitations sections, that the lack of weight-based dosing is a major constraint on the interpretability and generalizability of the results. The categories are center-specific constructs.

Incorporate Safety Data: The study is incomplete without an analysis of nephrotoxicity. The authors must go back to the data, apply a standard AKI definition (e.g., KDIGO), and report the incidence of AKI across the dosing groups. If the data is truly unavailable, this must be stated as a critical limitation that precludes any conclusions about the safety of lower doses.

Improve Statistical Reporting:

Remove the post-hoc power calculation.

Provide a table of post-matching covariate balances with Standardized Mean Differences (SMDs) to prove the effectiveness of the PSM.

Ensure all p-values and confidence intervals are reported consistently.

Refine the Discussion and Conclusions:

Tone down definitive language. Use "associated with" instead of "caused" or "led to."

Remove any claim of "non-inferiority" or "fewer adverse events" unless supported by data.

Frame the primary finding (higher mortality with usual dose) within the context of "confounding by indication" as the most plausible explanation.

Thorough Language and Copy-Editing: A native English speaker or professional editing service should review the entire manuscript to correct grammatical errors, improve sentence flow, and ensure consistent terminology.

Final Verdict: The manuscript reports a valuable and interesting real-world analysis. With the major revisions outlined above particularly a more critical acknowledgment of its limitations and the inclusion of safety data—it has the potential to be a significant contribution to the literature. In its current form, it is not ready for publication but is a strong candidate for major revision.

Reviewer #3: Overall, the authors have addressed my major methodological and interpretative concerns and substantially improved the manuscript. The handling of missing data, exposure categorization, dialysis subgroups, and unmeasured confounding is now described more transparently, and the statistical methods (including reporting of confidence intervals and justification of the propensity-score caliper) are clearer and more appropriate. The Discussion has been reframed with a more cautious, hypothesis-generating tone, with better integration of recent PK/TDM literature, and the limitations around MIC data, toxicity assessment, and single-centre generalizability are now explicitly acknowledged. The abstract now reports group-specific mortality, tables/figures are clearer, terminology is more consistent, and the reference on MDR definitions has been added. I would recommend acceptance for this manuscript.

**Do you want your identity to be public for this peer review?** For information about this choice, including consent withdrawal, please see our Privacy Policy

Reviewer #1: **Yes:** Dr Hammad Ahmed

Reviewer #3: **Yes:** SALMAN ASHFAQ AHMAD

---

## [Author Response · Author response to Decision Letter 2]

29 Dec 2025

Reviewer #1:

Comment 1:

Address the Fixed-Dosing Limitation: This is the top priority. The manuscript must explicitly state, in both the methods and limitations sections, that the lack of weight-based dosing is a major constraint on the interpretability and generalizability of the results. The categories are center-specific constructs.

Response:

We thank the reviewer for highlighting this critical issue. We have now explicitly acknowledged the absence of weight-based dosing as a major methodological limitation affecting both interpretability and generalizability. In the methods section, we clarify that polymyxin B dosing categories were derived from centre-specific fixed-dose prescribing practices, driven by routine documentation in lakh units and the unavailability of reliable body weight data in critically ill ICU patients. We emphasize that these categories represent real-world prescribing constructs rather than pharmacologically validated, weight-based regimens.

In the limitations section, we further highlight that the inability to normalize doses on a mg/kg basis precludes direct comparison with guideline-recommended dosing strategies and may introduce exposure misclassification. This constraint is now clearly stated as a major limitation of the study and reinforces that our findings should be interpreted as hypothesis-generating rather than practice-changing.

Methods: Polymyxin B dosing in our centre is routinely prescribed and documented in fixed lakh-unit doses rather than as weight-based regimens. Accurate body weight was not consistently available for critically ill ICU patients due to clinical instability and logistical constraints, precluding recalculation of doses on a mg/kg basis. Consequently, dosing categories (‘low’, ‘usual’ and ‘high’) were derived from centre-specific prescribing patterns and reflect real-world practice (1) rather than guideline-recommended, weight-adjusted dosing strategies. This approach captures actual clinical decision-making in our setting but limits direct pharmacokinetic interpretation and comparison with international dosing recommendations.

Limitations: An important limitation of this study is the absence of body weight for a substantial proportion of patients, which prevented normalization of polymyxin B dosing on a mg/kg basis. As a result, the fixed-dose categories used in this analysis represent centre-specific prescribing constructs rather than pharmacologically standardized dosing thresholds. This limitation restricts the interpretability of dose-response relationships and limits the generalizability of our findings to settings where weight-based dosing is routinely implemented. Accordingly, the observed associations should be interpreted cautiously and viewed as hypothesis-generating.

Comment 2:

Incorporate Safety Data: The study is incomplete without an analysis of nephrotoxicity. The authors must go back to the data, apply a standard AKI definition (e.g., KDIGO), and report the incidence of AKI across the dosing groups. If the data is truly unavailable, this must be stated as a critical limitation that precludes any conclusions about the safety of lower doses.

Response:

We fully agree with the reviewer that nephrotoxicity is a key safety outcome when evaluating polymyxin B dosing strategies. We carefully re-evaluated the available dataset to determine whether a standardized AKI definition could be applied. However, KDIGO-based AKI classification could not be reliably performed because serial serum creatinine values and urine output measurements were inconsistently documented at the required time points, and many patients initiated renal replacement therapy early during their ICU course.

Consequently, while AKI events were recorded as part of routine clinical documentation, these could not be adjudicated using standardized criteria nor casually attributed to polymyxin B exposure versus sepsis-related organ dysfunction or hemodynamic instability. We have therefore explicitly stated this as a critical limitation in the methods and limitations sections and clarified that the absence of standardized nephrotoxicity assessment precludes any conclusions regarding the safety or toxicity advantages of lower-dose regimens. The manuscript has been revised accordingly to avoid any safety-related inference.

Methods: Although AKI events were extracted from clinical records, standardized adjudication using KDIGO criteria was not feasible because serial serum creatinine measurements and urine output data were inconsistently available at required time points, particularly in patients with early initiation of renal replacement therapy.

Results: These AKI events reflect clinician documented diagnoses rather than standardized KDIGO-defined nephrotoxicity and should therefore be interpreted descriptively rather than as a formal safety comparison across dosing groups.

Limitations: AKI events, though documented in medical records, could not be reliably attributed to polymyxin B exposure versus sepsis-related organ dysfunction or hemodynamic instability. This precludes any conclusions regarding the comparative safety or nephrotoxicity advantage of lower versus higher polymyxin B dosing regimens and our findings should not be interpreted as evidence of reduced toxicity.

Comment 3:

Improve Statistical Reporting:

i. Remove the post-hoc power calculation.

Response:

We thank the reviewers for this suggestion. All post-hoc power calculations have been completely removed from the manuscript. Consistent with best statistical practice, we now focus on reporting effect estimates with corresponding 95% confidence intervals to convey the magnitude and precision of the observed associations.

ii. Provide a table of post-matching covariate balances with Standardized Mean Differences (SMDs) to prove the effectiveness of the PSM.

Response:

We thank the reviewer for this important suggestion. To formally demonstrate the effectiveness of the propensity score matching, we have now added a dedicated table reporting standardized mean differences (SMDs) for all covariates included in the propensity score model after matching. This table has been included as supplementary table II. Following matching, all covariates achieved acceptable balance with SMDs below 0.1, indicating adequate covariate balance across dosing groups.

iii. Ensure all p-values and confidence intervals are reported consistently.

Response:

We thank the reviewer for this comment. We have carefully reviewed the entire manuscript and standardized the reporting of all p-values and confidence intervals. P-values are now reported consistently using lowercase “p” and effect estimates (hazard ratios and odds ratios) are presented with corresponding 95% confidence intervals in a uniform format throughout the text, tables and figures. These revisions have been applied across the abstract, results, tables and supplementary materials.

Comment 4:

Refine the Discussion and Conclusions:

i. Tone down definitive language. Use "associated with" instead of "caused" or "led to."

Response:

We thank the reviewer for this important interpretative suggestion. We have revised the discussion and conclusions to remove definitive or causal language throughout the manuscript. All statements implying causation (eg., “caused”, “led to”, “resulted in”) have been replaced with associative terminology (eg., “was associated with”, “was observed with”, “correlated with”), consistent with the observational nature of the study. The conclusions have been correspondingly softened to emphasize associations rather than causality.

ii. Remove any claim of "non-inferiority" or "fewer adverse events" unless supported by data.

Response:

We agree with the reviewer and have removed all claims implying non-inferiority or reduced adverse events. The manuscript has been carefully revised to eliminate terms such as “non-inferior”, “safer” and “fewer adverse events”, as formal non-inferiority testing and standardized toxicity assessments were not performed. All statements have been reframed to describe observed associations only, without implying equivalence or safety advantages.

iii. Frame the primary finding (higher mortality with usual dose) within the context of "confounding by indication" as the most plausible explanation.

Response:

We thank the reviewer for this important interpretative point. We have now explicitly framed the observed association between usual dose polymyxin B and higher mortality within the context of confounding by indication. In the discussion, we clarify that clinicians may have preferentially prescribed usual or higher doses to patients perceived as more severely ill or clinically deteriorating, based on factors not fully captured by APACHE II or SOFA scores. We emphasize that this residual confounding is the most plausible explanation for the observed association and that the findings should be interpreted as hypothesis-generating rather than indicative of a true causal or dose-dependent effect.

Discussion: The finding of higher 28-day mortality among patients receiving the usual-dose polymyxin B regimen should be interpreted with caution. Confounding by indication is the most plausible explanation for this association. In routine clinical practice, clinicians may escalate polymyxin B dosing in patients perceived to have more severe illness, poor early response, or unfavourable prognostic features that are not fully captured by severity scores such as APACHE II or SOFA. Although propensity score matching was used to balance measured confounders, residual confounding from unmeasured clinical factors likely remains. Therefore, the observed association should not be interpreted as evidence of a harmful effect of usual-dose therapy but rather as a reflection of treatment selection bias inherent to retrospective observational studies.

Comment 5:

Thorough Language and Copy-Editing: A native English speaker or professional editing service should review the entire manuscript to correct grammatical errors, improve sentence flow, and ensure consistent terminology.

Response:

We thank the reviewers for this suggestion. The manuscript has undergone comprehensive language editing to improve grammatical accuracy, sentence structure, clarity and consistency of terminology throughout the text, tables, figures and supplementary materials. Revisions were applied across all sections to enhance readability and ensure alignment with journal standards.

Final Verdict: The manuscript reports a valuable and interesting real-world analysis. With the major revisions outlined above particularly a more critical acknowledgment of its limitations and the inclusion of safety data—it has the potential to be a significant contribution to the literature. In its current form, it is not ready for publication but is a strong candidate for major revision.

Response:

We sincerely thank the reviewer for their thoughtful evaluation of our work and for recognizing the potential contribution of this real-world analysis. We have carefully addressed all major concerns raised, including a more critical and explicit acknowledgment of the study’s limitations, removal of unsupported interpretative claims, refinement of causal language, and transparent clarification regarding the unavailability of standardized safety (nephrotoxicity) assessment. We believe that the extensive revisions made in response to the reviewer’s detailed comments have substantially strengthened the methodological rigor, interpretative clarity, and overall quality of the manuscript.

Reviewer #3:

Overall, the authors have addressed my major methodological and interpretative concerns and substantially improved the manuscript. The handling of missing data, exposure categorization, dialysis subgroups, and unmeasured confounding is now described more transparently, and the statistical methods (including reporting of confidence intervals and justification of the propensity-score caliper) are clearer and more appropriate. The Discussion has been reframed with a more cautious, hypothesis-generating tone, with better integration of recent PK/TDM literature, and the limitations around MIC data, toxicity assessment, and single-centre generalizability are now explicitly acknowledged. The abstract now reports group-specific mortality, tables/figures are clearer, terminology is more consistent, and the reference on MDR definitions has been added. I would recommend acceptance for this manuscript.

Response:

We sincerely thank the reviewer for their careful reassessment of the manuscript and for the positive and constructive feedback. We are pleased that the revisions have addressed the methodological and interpretative concerns, and that the improvements in transparency, statistical reporting, discussion framing, and acknowledgment of limitations were found to be appropriate. We appreciate the reviewer’s time and consideration and have retained these revisions in the current version of the manuscript.

References

1. Liu S, Wu Y, Qi S, Shao H, Feng M, Xing L, et al. Polymyxin B therapy based on therapeutic drug monitoring in carbapenem-resistant organisms sepsis: the PMB-CROS randomized clinical trial. Crit Care. 2023 June 13;27(1):232.

---

## [Decision Letter · Decision Letter 2]

29 Jan 2026

Clinical outcomes with lower versus conventional dose polymyxin B regimens in dialysis dependent and non-dialysis patients with gram-negative sepsis: A real-world propensity-score matched cohort study

PONE-D-25-38993R2

Dear Dr. Thunga,

We’re pleased to inform you that your manuscript has been judged scientifically suitable for publication and will be formally accepted for publication once it meets all outstanding technical requirements.

Kind regards,

Saswat Mohapatra, Ph.D.

Academic Editor

PLOS One

Additional Editor Comments (optional):

Reviewers' comments:

Reviewer's Responses to Questions

**Comments to the Author**

Reviewer #3: All comments have been addressed

2. Is the manuscript technically sound, and do the data support the conclusions?

Reviewer #3: Yes

3. Has the statistical analysis been performed appropriately and rigorously?

Reviewer #3: Yes

4. Have the authors made all data underlying the findings in their manuscript fully available?

Reviewer #3: Yes

5. Is the manuscript presented in an intelligible fashion and written in standard English?

Reviewer #3: Yes

Reviewer #3: The authors have satisfactorily addressed all major concerns raised in the previous round of review. The revised manuscript now provides a transparent and methodologically sound account of exposure classification, missing data handling, dialysis subgroup analyses, and residual confounding. Statistical reporting has been substantially strengthened, with consistent use of confidence intervals, standardized mean differences to demonstrate covariate balance, and clear justification of the propensity-score caliper. The Discussion has been appropriately reframed with a cautious, hypothesis-generating tone, explicitly acknowledging key limitations related to MIC availability, nephrotoxicity assessment, fixed-dose exposure misclassification, and single-centre generalizability. Importantly, the authors now contextualize the mortality findings within the framework of confounding by indication rather than causal inference. The abstract, tables, figures, terminology, and references have also been improved for clarity and consistency. Overall, the revisions have significantly enhanced the rigor, transparency, and interpretability of the study, and I support its acceptance in the current form.

**Do you want your identity to be public for this peer review?** For information about this choice, including consent withdrawal, please see our Privacy Policy

Reviewer #3: **Yes:** Salman Ashfaq Ahmad

---

## [Editor Report · Acceptance letter]

PONE-D-25-38993R2

PLOS One

Dear Dr. Thunga,

I'm pleased to inform you that your manuscript has been deemed suitable for publication in PLOS One. Congratulations! Your manuscript is now being handed over to our production team.

Kind regards,

on behalf of

Dr. Saswat Mohapatra

Academic Editor

PLOS One